Data aggregation algorithm for wireless sensor networks with different initial energy of nodes

Liu Zhenpeng 1 2
Zhang Jialiang 1
Liu Yi 2
Feng Fan 2
Liu Yifan 1 lyf@hbu.edu.cn
1 School of Cyber Security and Computer, Hebei University , Baoding, Hebei , China
2 Information Technology Center, Hebei University , Baoding, Hebei , China
Alatas Bilal
Electronic publication date: 2024 Mar 15
Publication date: 2024
Volume: 10
Electronic Location ID: e1932
Received 2023 Nov 16; Accepted 2024 Feb 16
Copyright: © 2024 Liu et al.
Copyright year: 2024
Copyright holder: Liu et al.
License: This is an open access article distributed under the terms of the Creative Commons Attribution License, which permits unrestricted use, distribution, reproduction and adaptation in any medium and for any purpose provided that it is properly attributed. For attribution, the original author(s), title, publication source (PeerJ Computer Science) and either DOI or URL of the article must be cited.
License URL: https://creativecommons.org/licenses/by/4.0/

Keywords: Data aggregation, Different initial energy, Privacy-preserving, Slice-and-mix technology, Wireless sensor networks, Node death rate, Tree topology, Communication overhead, Energy consumption, Dynamically reconfigure

Funding: National Natural Science Foundation of Hebei Province, China F2019201427 Fund for Integration of Cloud Computing and Big Data, Innovation of Science and Education (FII) of Ministry of Education of China 2017A20004 This research was supported by the National Natural Science Foundation of Hebei Province, China under Grant No. F2019201427 and the Fund for Integration of Cloud Computing and Big Data, Innovation of Science and Education (FII) of the Ministry of Education of China under Grant No. 2017A20004. The funders had no role in study design, data collection and analysis, decision to publish, or preparation of the manuscript.

==============================
Data aggregation plays a critical role in sensor networks for efficient data collection. However, the assumption of uniform initial energy levels among sensors in existing algorithms is unrealistic in practical production applications. This discrepancy in initial energy levels significantly impacts data aggregation in sensor networks. To address this issue, we propose Data Aggregation with Different Initial Energy (DADIE), a novel algorithm that aims to enhance energy-saving, privacy-preserving efficiency, and reduce node death rates in sensor networks with varying initial energy nodes. DADIE considers the transmission distance between nodes and their initial energy levels when forming the network topology, while also limiting the number of child nodes. Furthermore, DADIE reconstructs the aggregation tree before each round of data transmission. This allows nodes closer to the receiving end with higher initial energy to undertake more data aggregation and transmission tasks while limiting energy consumption. As a result, DADIE effectively reduces the node death rate and improves the efficiency of data transmission throughout the network. To enhance network security, DADIE establishes secure transmission channels between transmission nodes prior to data transmission, and it employs slice-and-mix technology within the network. Our experimental simulations demonstrate that the proposed DADIE algorithm effectively resolves the data aggregation challenges in sensor networks with varying initial energy nodes. It achieves 5–20% lower communication overhead and energy consumption, 10–20% higher security, and 10–30% lower node mortality than existing algorithms.

Introduction

Wireless sensor networks (WSNs) are computer networks consisting of numerous automatic devices distributed across various spatial locations. These devices employ sensors to detect information from different locations, such as temperature, humidity, and wind speed. The devices then create a connected network topology that transfers the collected information to a processing center in a multi-hop fashion. This enables the network to monitor the condition of the destination area in real-time using a collaborative approach with nodes (Gulati et al., 2022; Guleria & Verma, 2019; Mittal, Singh & Sohi, 2019; Sahoo, Pandey & Amgoth, 2021). WSNs have been widely used in various fields, including military, medical, and agricultural applications. In the military, WSNs are used for surveillance and reconnaissance operations. In medical applications, they are used to monitor patients’ health conditions in real-time. In agricultural fields, WSNs are utilized to monitor crop conditions and optimize irrigation and fertilization processes (Ali et al., 2020; Thakur et al., 2019; Tarannum & Farheen, 2020).

WSNs face various challenges in practical production applications due to limitations in energy, computing power, storage space, and other conditions of the sensor nodes (Kandris et al., 2020). Additionally, the natural characteristics of the regional distribution of sensor nodes further contribute to these challenges. Issues such as poor privacy of sensor data and uneven energy consumption are prevalent and pose significant challenges in the practical application of WSNs. Data aggregation emerges as a key method to address these challenges, as it can improve data privacy and energy efficiency in WSNs (Abbasian Dehkordi et al., 2020). Numerous studies have focused on developing and improving data aggregation algorithms in wireless sensor networks. However, in practical scenarios, there can be significant variations in the initial energy levels of sensor nodes. Some nodes may have low initial energy, but participate in the excessive data aggregation, while others with high initial energy may have fewer data aggregation tasks (Sharmin, Ahmedy & Md Noor, 2023). This energy imbalance can lead to premature death of certain nodes in the network topology, while others remain underutilized. Such situations have negative consequences on data aggregation and transmission in the entire network, resulting in energy consumption and communication overhead exceeding expectations (Rose & Ameelia Roseline, 2023). Moreover, it increases the node death rate, undermining effective load balancing. Addressing the issue of energy imbalance in WSNs with varying initial energy nodes is critical to improving the overall performance of WSNs. In WSNs with nodes of varying initial energies, addressing the energy imbalance problem is crucial for enhancing the overall performance of the networks. The challenges faced by WSNs in real-world production applications, such as energy constraints, unbalanced data aggregation, high node mortality rates, and the impact of nodes with different initial energies on data aggregation, highlight the need to develop strategies that effectively tackle the energy imbalance among sensor nodes. By addressing this issue, the overall performance of WSNs can be significantly improved.

To address the challenges mentioned earlier, this article proposes a novel data aggregation algorithm called Data Aggregation with Different Initial Energies (DADIE). The algorithm aims to enhance privacy protection, reduce energy consumption, and decrease the node death rate in wireless sensor networks (WSNs) where each node has a different initial energy level. DADIE consists of three phases: the topology formation phase, slicing and mixing phase, and data transmission phase. In the topology formation phase, DADIE constructs a tree topology for the sensor network, considering both energy and transmission distance (Kong et al., 2019). The algorithm takes into account that nodes closer to the base station (BS) tend to receive, process, and transmit more data, leading to increased energy consumption and premature death of these nodes. By considering energy consumption for data transmission between nodes, DADIE constructs the topology of the sensor network as a tree, taking into account energy and transmission distance. When a node in the topology tree selects a child node, it prefers the node with a small distance and large energy as its child node. This ensures that nodes with higher energy levels become intermediate nodes in the topology tree and handle more data reception and transmission. To prevent premature energy exhaustion of intermediate nodes, DADIE limits the number of child nodes for each intermediate node and dynamically reconstructs the aggregation tree before each round of data transfer, aiming to reduce the death rate of intermediate nodes. Moreover, DADIE establishes secure transmission channels between any two nodes to minimize the risk of data loss (Günther, 2020). In the slicing and mixing phase, which is the second phase of DADIE, the leaf nodes of the constructed topology tree employ the slice-and-mix technique. This technique involves slicing and randomly transmitting the data of the leaf nodes to adjacent nodes. Each node, then mixes the received data slices with its own sensory data to create its transmission data, ensuring the security of data aggregation. Additionally, the aggregation tree reconfiguration operations enhance the security of the network. Finally, in the data transmission phase, the transmission process starts from the leaf nodes. Each leaf node transfers its own data to the parent node. The parent node receives data from all its child nodes, performs mixing and processing, and passes the aggregated data to its parent node. This process continues until all the data reaches the BS. The proposed DADIE algorithm effectively addresses privacy protection, energy consumption, and node death rate in WSNs with varying initial energy levels. The three phases of DADIE, namely topology formation, slice mixing, and data transmission, work together to optimize data aggregation while considering energy efficiency and security aspects.

In this study, we propose a data aggregation algorithm called DADIE for WSNs in scenarios where the initial energy of each node in the network is different. The main contributions of this article are as follows: In the topology formation stage of the sensor network, a tree topology is employed, where the selection of intermediate and leaf nodes is based on their transmission distance and initial energy levels. Furthermore, to prevent the premature death of intermediate nodes, a restriction is imposed on the number of child nodes allowed for each intermediate node, and the aggregation tree is reconstructed before each round of data transmission. This approach aims to achieve a balance between energy consumption and the lifespan of data aggregation within the sensor network.

To ensure security in data aggregation, we establish secure transmission channels between nodes involved in data transmission. Additionally, at the leaf nodes, we employ slice-and-mix techniques prior to data transmission.

We conducted simulation experiments using MATLAB to compare the DADIE algorithm with the proposed PECDA algorithm (Wang et al., 2018) and EPDA algorithm (Zhou et al., 2019) which perform WNSs scenario and sensor node parameters provided by the Intel Berkeley Research Lab, as referenced in Samuel (2004). The results demonstrate that DADIE outperforms these two algorithms in terms of data privacy, communication overhead, energy consumption, and node death rate. Specifically, compared to the PECDA and EPDA algorithms, DADIE improves data privacy by 10–20%. It also reduces communication overhead and energy consumption by 5–20%. Furthermore, DADIE achieves a significant reduction in the node death rate of approximately 10–30%.

The remainder of this article is organized as follows: “Related Work” provides a review of related research on the DADIE algorithm. In “Preliminary Work”, we describe the preliminary preparation required for implementing the DADIE algorithm. “DADIE Algorithm” introduces the specific implementation steps of the DADIE algorithm. “Performance Analytics” focuses on the performance analysis of the DADIE algorithm. In “Simulation and Analysis”, we conduct a comparative experiment. Finally, in “Conclusion”, we conclude the article and provide an outlook on future research directions.

Related work

Data aggregation plays a vital role in enhancing data privacy, conserving network energy, and extending the network lifetime during data transmission in wireless sensor networks (WSNs). Numerous previous studies have proposed various data aggregation algorithms, which can be broadly classified into two categories: (1) Data aggregation algorithms based on tree topology, and (2) Data aggregation algorithms based on cluster topology.

Tree topology

The SMART algorithm, proposed by He et al. (2007) serves as a data privacy-enhancing solution for data aggregation operations in tree topology. SMART utilizes a tree-topology-based data slice-and-mix transmission algorithm, which involves three steps: slicing, mixing, and data transfer. During the slicing stage, each node divides its data into J segments, retains one segment, and randomly transmits the remaining J-1 segments to its J-1 neighboring nodes. In the mixing phase, each node combines the received data fragments with its own data to generate its transmission data. Finally, during the data transfer phase, all nodes transmit their mixed data to the parent node and eventually to the receiver. Another algorithm, PECDA, was proposed by Wang et al. (2018) as a means of achieving privacy-preserving and energy-efficient continuous data aggregation in wireless sensor networks. PECDA is based on the SMART algorithm (Wang et al., 2018). PECDA employs the slice-and-mix technique proposed in SMART on leaf nodes in the network topology, while non-leaf nodes establish secure channels with neighboring nodes to ensure data privacy. Additionally, PECDA takes into account the temporal correlation of data sensed by leaf nodes during continuous data aggregation to reduce the communication overhead in the network and extend the network lifetime. However, it is important to note that this approach may increase the probability of data leakage in some nodes. Hajian & Erfani (2021) proposed a continuous data aggregation algorithm for wireless sensor networks that considers the spatial correlation of nodes. This algorithm takes into account data similarity resulting from the spatial correlation of node locations. Multiple nodes with high correlation alternate between active and sleep states, reducing communication overhead and energy consumption in the network. Additionally, the algorithm employs data slicing and mixing between nodes in sensor networks to enhance privacy protection.

The EPDA chain-based data aggregation algorithm for wireless sensor networks was proposed by Zhou et al. (2019). EPDA elongates sub-trees in the network to form a chain structure, reducing the number of leaf nodes and data slices, thereby achieving a reduction in communication overhead. However, the chained structure increases the number of hops for nodes to reach the receiver, resulting in increased communication overhead for data transmission from trailing nodes to the receiver. Consequently, the chained structure leads to an unbalanced energy load in the network, and some intermediate nodes close to the receiver may experience premature death. Dao et al. (2023) proposed a heuristic algorithm called RRPT to reduce redundant data packets in wireless sensor networks, aiming to maximize their lifespan by minimizing the total transmission and reception energy consumption of sensor nodes. The algorithm constructs an aggregation tree with as few redundant data packets as possible. When a new node joins the aggregation tree, the selection of its parent node is based on the minimum number of redundant data packets along the path from the parent node to the receiver. However, the algorithm does not consider the hierarchical structure of the tree, which can result in excessive levels of the tree structure and higher energy consumption during data transmission. In another study, Tai et al. (2023) consider the known topology of the sensor network and propose a power control strategy for each node based on fairness and end-to-end Quality of Service (QoS) constraints. The objective is to reduce node energy consumption and prolong the overall network lifespan. The optimal strategy computation employs the Lagrange relaxation method as a solution approach. By decomposing the problem into directly solvable subproblems based on different decision variables, each iteration yields an approximately optimal feasible solution until the optimal solution is obtained. This method takes into account the optimal strategy of sensor nodes under practical constraints. However, it does not consider the limitations imposed by the tree topology structure on sensor nodes and its impact on data collection.

Cluster topology

A cluster-based data aggregation algorithm for wireless sensor networks was proposed by He et al. (2007). In this algorithm, WSN nodes are organized into clusters of varying sizes, and cluster heads are selected within each cluster to perform intra-cluster data aggregation. The aggregation results are then transmitted to the receiver. Heinzelman, Chandrakasan & Balakrishnan (2000) proposed the LEACH aggregation algorithm for wireless sensor networkss. This algorithm dynamically selects cluster heads to achieve energy-balanced loads during cluster data aggregation. LEACH enables nodes to form clusters of different sizes and selects cluster heads for intra-cluster data aggregation. In another work (Sekar, Suganthi & Dheepa, 2023), a routing protocol called JSARP is proposed to optimize the selection of cluster head nodes and reduce network energy consumption to improve the network lifespan. JSARP selects cluster heads based on relevant fitness attributes. Once the cluster heads are determined, the jellyfish swarm algorithm is used to choose the optimal path for data transmission, aiming to minimize energy consumption in data collection and prolong the network’s lifespan. However, this method solely focuses on the optimal path for data transmission and does not consider the imbalance in energy consumption across the network. Consequently, the overall energy consumption of the network may not be evenly distributed.

In this study, we propose the DADIE algorithm for data aggregation in wireless sensor networks, specifically designed to handle scenarios where network nodes have different initial energy levels. By leveraging the aforementioned data aggregation algorithm, DADIE aims to decrease network communication overhead and energy consumption, while enhancing network privacy protection and extending the network’s lifetime. In comparison to existing data aggregation algorithms, DADIE specifically addresses the case of data aggregation with WSN nodes having varying initial energy levels. It offers improvements over existing algorithms by effectively dealing with the challenges posed by an unbalanced energy load.

Preliminary work

In this section, we begin by introducing the models that form the foundation of the DADIE algorithm. These models include the network model and the attack model, which are essential prerequisites for implementing DADIE.

Network model

In this article, the wireless sensor network (WSN) comprises a receiver, also known as the base station (BS), and N sensor nodes. All nodes are deployed within an area of size M*M and form a connected network denoted as G (V, E), where V represents the number of nodes in the sensor network structure and E represents the set of links present in the sensor network G. In this network, all nodes have an identical data transmission radius, denoted as R. We assume the following conditions: In the wireless sensor network, all nodes are uniformly and randomly distributed throughout the network area, with fixed positions. Each node is assigned a unique identifier (ID) and possesses a distinct initial energy value. These nodes have the capability to sense data from the environment, receive data from neighboring nodes, and transmit data to other nodes within their transmission radius, denoted as R.

The network G is composed of three types of nodes: the data receiver BS, intermediate nodes, and leaf nodes. The data receiver BS is equipped with unlimited energy, computing power, and storage space. Therefore, we do not consider its energy consumption and communication overhead in our analysis. The intermediate nodes in the network receive data from their child nodes, aggregate the received data with their own data, and generate a packet that is transmitted to their respective parent node. On the other hand, the leaf nodes in the network sense data from the environment, perform slice-and-mix operations on the sensed data, and then transmit the processed data to their parent node.

Neighboring nodes in the network establish secure data transmission channels to ensure the confidentiality and integrity of the transmitted data. These secure channels are used for all data transmissions within the network, ensuring that all data is transmitted securely.

In this study, the data aggregation performed by each node is referred to as complete aggregation. This means that the node retains all the data it receives and encapsulates it into a single packet for transmission. Specifically, we adopt a summation aggregation approach in this article, where the received data at a node is aggregated by performing an additive summation operation.

Before each cycle of data transmission, the nodes in the network complete the data sensing process. This ensures that the nodes have gathered the required data before initiating the transmission. It is important to note that the data sending and receiving operations of a node cannot be performed simultaneously. In other words, a node cannot send and receive data at the same time. These operations occur sequentially to ensure proper data transmission and reception within the network.

Attack model

Due to the specific characteristics of node distribution and data transmission in sensor networks, attackers primarily launch attacks from two directions (Yousefpoor et al., 2021; Salmi & Oughdir, 2023; Cao et al., 2022): Eavesdropping attacks are a prevalent form of attack in wireless sensor networks. In WSNs, where data transmission occurs wirelessly between nodes, attackers can intercept and eavesdrop on the communication channel. By doing so, they can capture the link information exchanged between the transmitting node and the receiving node.

Cracking attacks represent another type of attack in wireless sensor networks. In this form of attack, the attacker gains unauthorized access to a sensor node and obtains various sensitive information, such as data, keys, location details, and other pertinent data related to the compromised node. Furthermore, through the compromise of one or multiple sensor nodes, the attacker may extract information about neighboring nodes, resulting in a significant breach of the entire network’s security.

Dadie algorithm

Based on the identified challenges associated with wireless sensor networks comprising nodes with different initial energy levels, we propose a novel data aggregation algorithm called Data Aggregation with Different Initial Energies (DADIE). The primary objective of this algorithm is to facilitate efficient data aggregation in such networks. This section outlines the specific implementation details of the DADIE algorithm.

Basic idea of DADIE

The DADIE algorithm is specifically designed for data aggregation in scenarios where the initial energy levels of nodes in a sensor network vary. The algorithm consists of four key components: Initialization phase: In the initialization phase of the DADIE algorithm, pre-allocated keys are stored in the memory of the sensor nodes before their deployment. Once the nodes are deployed, they begin sensing and collecting information regarding their location, initial energy levels, and other relevant data.

Topology formation phase: During the topology formation phase of the DADIE algorithm, the goal is to create an aggregation tree that establishes a tree-like structure for the wireless sensor network. The BS serves as the root node, and other nodes are added to the tree starting from the BS. The DADIE algorithm considers both distance and energy factors to optimize the construction process. To account for the higher energy consumption of intermediate nodes closer to the BS, which occurs due to increased data collection, processing, and transmission responsibilities, the DADIE algorithm employs a cyclic approach. It selects nodes that have not yet joined the aggregation tree but possess the minimum distance and maximum initial energy. These selected nodes are given priority to become intermediate nodes in the tree. However, to prevent excessive energy consumption and premature failure of intermediate nodes, the DADIE algorithm sets a constraint on the maximum number of child nodes allowed for each intermediate node. This constraint ensures a balanced energy consumption among the nodes in the aggregation tree. Furthermore, the aggregation tree is reconstructed before each round of data transmission to adapt to potential changes in node energy levels and network conditions. This dynamic reconstruction ensures the efficiency and adaptability of the tree structure, enabling better utilization of network resources and prolonging the network lifetime.

During the slicing and mixing phase of the DADIE algorithm, leaf nodes in the aggregation tree divide their sensory data into segments. One segment is retained as the node’s own data, while the remaining data slices are randomly transmitted to neighboring nodes. This process allows for data distribution and collaboration among the nodes. Each node receives data segments from its neighboring nodes. It then combines its own sensory data with the received data segments through a mixing process. This mixing operation allows the node to form the transmitted data that will be further propagated through the aggregation tree.

The data aggregation and transmission phase of the DADIE algorithm initiates with the leaf nodes in the aggregation tree. These leaf nodes transmit their data to their parent node, which is closer to BS. Upon receiving data from their child nodes, each parent node performs data aggregation by combining the received data with its own collected data. This aggregation process typically involves applying specific aggregation functions, such as averaging or summing, to the data. After the aggregation, the parent node transmits the aggregated data to its own parent node in the tree. This process of data aggregation and transmission continues recursively as the data moves up the aggregation tree towards the BS. Each intermediate node receives and aggregates data from its child nodes, forming a consolidated dataset. This aggregated data is then transmitted to the next higher-level node until it reaches the BS.

Description of DADIE

Establish secure channels

Due to the specific process of topology formation and the establishment of secure transmission channels between nodes, it is necessary to pre-assign keys to nodes before their deployment. This pre-assignment helps to reduce unnecessary communication overhead that would arise during key assignment while establishing secure transmission channels between neighboring nodes (Farjamnia, Gasimov & Kazimov, 2019; Jiang et al., 2022). In this study, we have adopted the key distribution scheme proposed in Eschenauer & Gligor (2002), which involves three steps: (1) Key pre-distribution phase: During this phase, a key pool is generated based on the size of the network. The optimal number of key rings to be stored in the memory of each undeployed sensor is calculated. These key rings contain pre-assigned keys that will be used for secure communication. (2) Shared-key discovery phase: In this phase, a node broadcasts the list of key identifiers present on its key ring. A link is established between two sensor nodes only if they share a common key. This shared-key discovery process allows for the identification of secure communication channels between nodes. (3) Secure transmission channel establishment phase: This phase involves the establishment of secure channels between sensor nodes. A secure channel exists between two nodes only if they share a common key. Furthermore, all communications on these secure channels are encrypted and protected to ensure data confidentiality and integrity.

In this study, we make the assumption that if the key ring size of two nodes is denoted as k, where the key ring is selected from a key pool of a given size P and assigned to the nodes, the probability of having at least one shared key between them is denoted as Pconnect′. Based on this assumption, we calculate the probability of a common key existing between two neighboring nodes as follows:

(1) Pconnect′=1−((P−k)!)2(P−2k)!P!.

Because there exists a possibility of node leaks, assume that the probability of a third party obtaining the leaked public key is Plost, then:

(2) Plost=kP.

In the DADIE algorithm, the establishment of secure transmission channels is crucial for ensuring the safe and reliable transmission of data and information among nodes. To achieve this, secure channels need to be established between nodes and their parents, as well as between neighboring nodes. To establish a secure channel between two neighboring nodes, let’s consider node A and node B. They utilize a pre-assigned secret key ring. The process starts with the sharing of a public key, denoted as K. Node A selects a random number, denoted as rAB, and encrypts it using the public key K. The resulting ciphertext is then transmitted to node B. Upon receiving the ciphertext, node B decrypts it using the public key K, obtaining the decryption result rBA. If the decrypted value rBA matches the original random number rAB, it indicates that both nodes share the same secret key. This matching verifies the authenticity and integrity of the communication, and confirms that a secure transmission channel can be established between nodes A and B. By utilizing these secure channels, the DADIE algorithm ensures that data transmission between nodes remains confidential and protected against eavesdropping attacks. The use of encryption prevents unauthorized access and maintains the integrity and privacy of the transmitted data, enhancing the overall security of the wireless sensor network.

Energy loss model

In this article, we utilize a first-order radio energy consumption model to analyze the energy consumption of the sensor nodes specifically during data transmission and reception. We focus solely on these aspects and do not consider the energy loss that may occur during computation and storage processes. The process is illustrated in Fig. 1, which is divided into two halves. The left half of the figure illustrates the energy consumption of the sensor node during the transmission of data, while the right half depicts the energy consumption during data reception (Gupta, Gupta & Verma, 2022). In Fig. 1, the energy consumption of a sensor node during data transmission is depicted. When a sensor node transmits N bits of data to another node, the energy consumed by the sending node can be calculated as N∗Een. Here, Een represents the energy consumed by the sensor node to transmit or receive 1 bit of data. After the data is transmitted, it passes through the power expansion circuit, which results in additional energy consumption. This additional energy consumption can be calculated as N∗β(L), where β(L) represents the amplification factor of the power expansion circuit for data transmission at different distances. Equations (3) and (4) are the calculations of β():

Figure 1 Flowchart of data transmission energy consumption.

(3) β(L)={εFS∗L2,L<L0εAMP∗L4,L≥L0

(4) L=(X1−X2)2+(Y1−Y2)2.

εFS and εAMP denote the energy expended in the receiver amplifier in the free space model and in the receiver amplifier in multipath fading model respectively (Eidaks et al., 2022; Tirmizi et al., 2022). Moreover, L0 denotes a distance threshold, L denotes the node communication distance and (X,Y) are the coordinates of the node. When the L is less than L0, the energy consumption is proportional to the square of the distance. On the other hand, when the L is greater than or equal to L0, the energy consumption is proportional to the fourth power of the distance (Hamad, Dag & Gucluoglu, 2023). The distance threshold is defined as:

(5) L0=εFSεAMP.

Thus the total energy consumed when a node sends N bit of data is:

(6) ENS=N∗Een+N∗β(L).

The energy consumed by a node to receive N bit of data is:

(7) ENR=N∗Een.

Form topology

In the DADIE algorithm, BS is designated as the root node of the aggregation tree, and the topology construction starts from the BS. Here is a description of the process: (1) BS broadcasting: The process begins with the BS broadcasting an “invite” message to nodes within its communication range. This message serves as an invitation to nearby nodes, encouraging them to establish a secure channel and become child nodes of the BS. (2) Node response: Upon receiving the invitation message, eligible nodes respond with an “agree” message to indicate their willingness to establish a secure transmission channel and become a child node. These nodes meet certain criteria or conditions specified by the algorithm to be considered eligible. (3) Information exchange: After agreeing to become child nodes, the responding nodes transmit their location, initial energy, and other relevant information to the BS. This data exchange allows the BS to gather necessary information about the nodes. (4) Parent-child relationship: Once the BS receives the “agree” message from a node and receives its transmitted information, it stores the node’s information and designates the corresponding node as a child node. The BS becomes the parent node of these child nodes in the constructed aggregation tree.

By following these steps, the DADIE algorithm constructs the topology of the wireless sensor network, with the BS as the root node and other nodes as child nodes. This hierarchical structure facilitates efficient data aggregation and communication throughout the network. The algorithm flow is shown in Algorithm 1 and Part 1 of Fig. 2A.

Algorithm 1 Child nodes of the BS join the topology tree.

  Data: Sensor Nodes Set N, Data Transmission Radius R	
1  set sizeof(Flag)=sizeof(N) and Flag= {0} as a flag to join the tree or not;	
2  set set of child nodes of BS, BSchild{};	
3  for eachNi∈N do	
4   BS sends “invite” message to Ni;	
5   if distance of BS and Ni<R and Flagi=0 then	
6    Ni reply “aggre” message to BS, set BS as parent node;	
7    BS set Ni as one of its child node;	
8    Establising a secure transmission channel between BS and Ni;	
9    Add node Ni to BSchild{};	
10    Flagi=1;	
11   end	
12  end	

Figure 2 Flowchart of DADIE algorithm: (A) the whole process of DADIE algorithm execution; (B) selection operation.

The distribution of nodes in a WSNs is depicted in Fig. 3A, along with the corresponding initial energy and position coordinates provided in Table 1. The BS, acting as the root node, is considered as the origin of the coordinate system (0,0). It initiates an “invite” message to the network, targeting nodes 1, 2, 3, and 4 located within its communication range. Upon receiving the “invite” message, these nodes respond with an “agree” message, providing their initial energy and coordinate position information to the BS. Additionally, they designate the BS as their parent node in the aggregation tree. Upon receiving this information, the BS stores the data and identifies these nodes as its child nodes. This process leads to the formation of the topology depicted in Fig. 3B, where the BS serves as the root node, and nodes 1, 2, 3, and 4 become its child nodes in the aggregation tree.

Figure 3 Tree network topology formation process: (A) the process of finding child nodes for BS nodes; (B) BS establishes connection with its child nodes; (C) initially the process of finding child nodes for the residual nodes; (D) the final process of finding child nodes for the residual nodes; (E) the resulting topology.

Table 1 Initial energy and position coordinate of nodes in the example.

Sensor nodes	Initial energy [J]	Position coordinate	
BS	∞	(0,0)	
1	1.42 J	(−4,2)	
2	0.92 J	(−2,3)	
3	1.52 J	(0.5,5)	
4	1.98 J	(4,3)	
5	1.62 J	(−6,5)	
6	1.51 J	(−3.5,6)	
7	1.68 J	(2.5,6)	
8	1.64 J	(5,6)	
9	1.70 J	(6,4)	

In the DADIE algorithm, intermediate nodes play a crucial role in sensing data, receiving data from leaf and child nodes, and aggregating the received data with their own data. They then transmit the aggregated result to their parent node. However, due to their increased responsibilities and data processing tasks, intermediate nodes, especially those closer to the BS, experience higher energy consumption, which can lead to premature node death. In order to achieve a balanced load of network energy and reduce node death rate as much as possible, make the initial energy value of intermediate nodes is as large as possible, also to make the data transmission distance of these nodes smaller. In addition, DADIE limits the number of children of each intermediate node and reassembles the aggregation tree before each round of data transfer. The algorithm is described as follows:

In DADIE, we introduce the parameter CHO, such that it serves as a measure of the node’s choice of child nodes; the parameter MAXchild is introduced such that it acts as the maximum number of child nodes of the intermediate node.

(8) CHO=EN/L

EN is the initial energy of the node to be added to the aggregation tree, L denotes the distance between the node to be added to the aggregation tree and this selection node. L uses Euclidean distance calculations, X and Y denote the coordinates of the node.

(9) MAXchild=random(1,Num)

Num indicates the number of nodes within communication range of this intermediate node that have not joined the aggregation tree. random(a,b) denotes a randomly generated integer between a and b (include a and b).

To achieve the aforementioned goals, the DADIE algorithm follows a specific procedure. Initially, the BS establishes a secure channel with the child nodes. Starting from the child node with the highest CHO of the BS, it is designated as the selection node. The selection node then broadcasts “pre_invite” messages throughout the network. Nodes that receive the “pre_invite” message and have not yet joined the aggregation tree respond by providing information about their location and initial energy level with the message “agree” to the selection node. Upon receiving these messages, the selection node counts the number of nodes that sent responses, denoted as Pnum. If Pnum is greater than or equal to 1, the selection node generates a random integer MAXchild within the range from 1 to Pnum, representing the maximum number of child nodes for a given node. The selection node then chooses the first MAXchild nodes with the highest CHO values as its children and sends an “invite” message to these nodes. Nodes that receive the “invite” message acknowledge the selection node as their parent and establish a secure communication channel with it. In the case where Pnum is equal to 0, indicating no responses were received, the selection node becomes a leaf node directly.

Once the selection node completes its selection operation, the node with the highest CHO value among its sibling nodes, who has not yet performed the selection operation, becomes the new selection node. If all sibling nodes have completed the selection operation, the process is repeated with the child node with the highest CHO value among the sibling nodes who performed the selection operation first. This process continues until all child nodes have completed the selection operation. If there are still nodes that have not been included in the aggregation tree, the node with the highest CHO value in the aggregation tree is selected as the parent node by these nodes. The algorithm flow is shown in Algorithm 2. Part 2 of Fig. 2A is the process by which nodes in the network other than the children of BS join the aggregation tree and Fig. 2B is the selection operation, and selection operation is the operation that every leaf node performs when it is looking for a child node.

Algorithm 2 Child nodes of remaining nodes join the topology tree.

1 ht	
  Data: Sensor Nodes Set N, Set of Child Nodes of BS BSchild{}, Flag={}, Node Communication Radius R, Maximum Number of Child Nodes, MAXchild, Initial Energy EN, Position Coordinate, X, Y, L, CHO	
2 set a queue Queue{0} as the nodes order for finding child nodes;	
3 set a set Flag_Queue= {0} as flag of a child node of BS is added to Queue{} or not;	
4 while each Flag_Queue= {} of BSchild{} !=1 do	
5  select the node in BSchild{} with the most CHO and Flag_Queue{} is 0 to join Queue{};	
6  set this node Flag_Queue{} to 1;	
7 end	
8 set the queue’s head pointer is head;	
9 set the queue’s tail pointer is tail;	
10   set Pnum=0;	
11   head and tail point to the head and tail of Queue{} respectively;	
12   while head≤tail do	
13   Pnum=0;	
14   Queue{head} sends “pre_invite” message to N;	
15  for eachNi∈N do	
16   if distance of Queue{head} and Ni≤ R and Flagi=0 then	
17    Ni reply “aggre” message with location and energy information to Queue{head};	
18    Pnum=Pnum+1;	
19   end	
20  end	
21  if Pnum>0 then	
22    MAXchild=random(1,Pnum);	
23   for eachNi∈NPnum do	
24     LNi=(XNi−XQueue{head})2+(YNi−YQueue{head})2;	
25     CHONi=ENNi/LNi;	
26    end	
27    Queue{head} sends “invite” message to top MAXchild of CHO in Ni;	
28   set these nodes Flagi=1;	
29   set these nodes into Queue{} by CHO;	
30   set these nodes Flag_Queue={} to 1;	
31   tail point to tail+1;	
32  end	
33  else	
34   head point to head+1;	
35  end	
36 end	

As shown in Figs. 3C and 3D, the BS computes the CHO for all its child nodes. Among the child nodes of the BS, node 4 has the largest CHO value. Node 4 then broadcasts a “pre_invite” message to the network. Nodes 7, 8, and 9, which are within the communication range of node 4, respond to the “pre_invite” message by sending an “agree” message that includes their position and initial energy information. Node 4 receives these “agree” messages and counts the number of received responses, denoted as Pnum. Since Pnum is greater than 1, node 4 generates a random integer MAXchild between 1 and Pnum. This value determines the maximum number of child nodes for node 4, which in this case is MAXchild=3. Subsequently, node 4 sends an “invite” message to nodes 7, 8, and 9, designating them as its child nodes and establishing a secure transmission channel with them. Nodes 7, 8, and 9 acknowledge node 4 as their parent and join the topology tree upon receiving the “invite” message. Once this operation is completed, the child node of the BS with the largest CHO value, which has not yet initiated the invitation operation, is node 1. Node 1 then starts the invitation operation by sending a “pre_invite” message to nodes 5 and 6 within its communication range. Nodes 5 and 6, which have not joined the aggregation tree, respond to node 1 with their location and initial energy information in an “agree” message. Node 1 counts the number of received “agree” messages, denoted as Pnum. Since Pnum is greater than 1, node 1 generates a random integer MAXchild between 1 and Pnum. In this case, MAXchild=1, indicating that node 1 can have at most one child node. Node 1 calculates the CHO values of nodes 5 and 6, and selects the node with the highest CHO value to send an “invite” message. It establishes a secure transmission channel with the selected node and designates it as its child node. Node 5, upon receiving the “invite” message from node 1, acknowledges node 1 as its parent and joins the topology tree. The above operations are repeated by node 2 and node 3 in sequence.

After all the child nodes of the BS have completed the above operation, among the child nodes of the node that first started the above operation among the child nodes of the BS, the node with the largest CHO is selected to repeat the above operation until all the nodes in the network have completed the above operation. The final topology is shown in Fig. 3E.

Slicing and mixing

To improve the privacy protection of the network and minimize the risk of data being intercepted by malicious attackers, it is recommended to slice-and-mix the data of leaf nodes before transmission. During this phase, all leaf nodes in the aggregation tree slice their sensory data into J (J ≥ 2) fragments. The leaf nodes retain one fragment as their transmission data and randomly transmit the remaining J-1 fragments to J-1 neighboring nodes (except for the parent node). The receiving node aggregates the received data fragments with its own data and employs the resulting aggregation as its transmission data. The value of J can either be pre-set in the node or broadcasted to each node in a flooded manner by the BS. The pre-set method reduces unnecessary communication consumption in the network but is less flexible, while the flooded method enhances flexibility but increases the network communication load. In DADIE, we opt for the pre-set method to establish the value of J. The execution process is as Part 3 of Fig. 2A.

As depicted in Fig. 4A, the number of fragments J is set to 3, and the leaf nodes in the network are node 5, 6, 3, 7, 8, and 9, f(j,i) denotes the size of the data fragment received by node i from node j. During this phase, each leaf node slices its data into three fragments, retaining one fragment as its transmission data and transmitting the other two fragments to neighboring nodes. For instance, node 5 transmits fragment f(5,6) to neighboring node 6 and fragment f(5,2) to node 2, and the remaining leaf nodes follow a similar protocol. It should be noted that if a leaf node receives a data fragment from another leaf node before its fragment transmission, the received data fragment will be directly incorporated into the leaf node’s data fragments, and the sensor data will be sliced into the residual number of fragments. For example, in Fig. 4B, node 6 has received fragment f(5,6) from node 5 before its fragment transmission, thus requiring node 6 to slice its data into two fragments, mix them with the received data fragment from node 5, and transmit two fragments to neighboring nodes 5 and 3. However, it is crucial to ensure that the fragment received by a node is not transmitted to the neighboring node that sent the fragment, to avoid data redundancy.

Figure 4 Flow of leaf nodes slice-and-mix operation: (A) fragments transmission operation of leaf nodes; (B) fragments mixing operation.

Following the completion of fragment transmission, the nodes in the aggregation tree aggregate the received data fragments with their own data, as demonstrated in Eq. (10) (where DNi denotes the transmission data of node i, k denotes the number of data fragments received by node i, and f(j,i) denotes the size of the data fragment received by node i from node j, ‘+’ denotes stitching data together), to generate the transmission data for that particular node, as illustrated in Fig. 4B.

(10) DNi=DNi+∑j=1kf(j,i)

Data transmission

During this phase, DADIE implements the transmission of data from the sensor network nodes to the BS. The leaf node of the aggregation tree transmits its aggregation result to the parent node through a secure transmission channel, and the intermediate node receives data from the child node, aggregates the received data with its own data to generate a new aggregation result, and subsequently transmits it to the parent node. The parent node repeats this operation until the data reaches the BS. As depicted in Fig. 5 and part 4 of Fig. 2A, nodes 7, 8, and 9 transmit the result of data mixing to their common parent node, node 4. Upon receiving data from child nodes, node 4 aggregates its data with data from child nodes and transmits the aggregation result to the parent node of node 4, i.e., the root node BS. The data volume of each node is calculated using Eq. (11) (where DNi denotes the transmission data of node i, m denotes the number of child nodes of node i, and DNj represents the volume of transmission data from node j to i.).

Figure 5 Data transmission stage.

(11) DNi=DNi+∑j=1mDNj.

Performance analytics

When evaluating data aggregation algorithms in wireless sensor networks (WSNs), several important metrics are considered to assess their effectiveness. Three key evaluation metrics are energy consumption, network lifetime, and security. 1) Energy consumption: Energy efficiency is a critical aspect to evaluate data aggregation algorithms in WSNs. The goal is to measure the relationship between energy consumption and the actual utilization of sensor nodes. By improving energy efficiency, the battery life of sensor nodes can be extended, leading to prolonged operational time for the entire network and enhancing its stability. Efficient data aggregation algorithms can significantly reduce the energy consumed during data transmission, processing, and communication.

2) Network lifetime: Network lifetime is another crucial metric to evaluate data aggregation algorithms. It directly relates to the sustainability and stability of the entire network. By optimizing energy consumption and minimizing unnecessary communication, data aggregation algorithms can effectively extend the overall lifespan of the network. Maximizing network lifetime is essential for applications that require long-term monitoring and data collection.

3) Security: Security is of utmost importance in evaluating data aggregation algorithms in wireless sensor networks. WSNs are prone to various security threats and attack risks (Guo, Liu & Liu, 2023). Evaluating algorithm security involves assessing the protection measures and defense mechanisms employed for data aggregation. The objective is to prevent vulnerabilities and mitigate the risks of attacks. Robust security measures ensure data integrity, confidentiality, and availability, thereby enhancing the trustworthiness and reliability of the transmitted data.

By considering energy efficiency, network lifetime, and security as evaluation metrics, researchers and practitioners can assess the effectiveness and suitability of different data aggregation algorithms in wireless sensor networks (Guruprasath, Nagarajan & Kannadhasan, 2023). These metrics help optimize energy consumption, extend the network’s lifespan, and ensure the protection and reliability of the data transmitted in the network.

We propose DADIE, a data aggregation method that integrates two parameters, energy and distance, to construct an aggregation tree, and in this section, we theoretically analyse the data privacy, energy consumption and communication overhead of the DADIE algorithm. Where the relevant parameters are shown in Table 2:

Table 2 Description of DADIE’s parameters.

Parameter	Description	
N	Number of nodes	
G	Nodes distribution area	
EN	Node initial energy	
R	Transmission radius	
J	Number of slices in leaf nodes	
Een	Energy consumed to send/receive 1 bit of data	
Ede	Energy consumed to process 1 bit of data	
Plost	The probability of a third party obtaining the leaked public key	
Pleaf	The probability of data leakage of leaf nodes	
Jrec	The number of data fragments received by this node	
Numfather	The number of parent nodes of this node and always equal to 1 energy	
Pinter	The probability of data leakage of the intermediate node	
Numchild	The number of child nodes of this intermediate node	
Nne	The number of neighboring nodes for each node	
Nrav	The average number of times each node replies to other nodes invitation messages	
Niav	The average number of invitation messages sent per node	
ρ¯	The invitation message is the same length as the reply message	
α	The proportion of leaf node	
D	Sensory data for each node	
DNIDS	Node information data size	
Dr	The total amount of data received by each node as data fragments	
Dchild	The total amount of data received by each intermediate node from the child nodes	

Security analysis

The data aggregation approach employed in this study utilizes the slice-and-mix technique and establishes secure transmission channels at nodes. It also reconstructs the aggregation tree before each round of data transfer, making it exceedingly difficult for an attacker to obtain all the data of a node in the network.

In one round of data aggregation in the DADIE algorithm, a leaf node partitions its data into J fragments. One fragment is retained by the leaf node, while the remaining fragments are transmitted to randomly selected neighboring nodes. Thus, the links of leaf nodes include both links for transmitting leaf node fragments and links between leaf nodes and their parents. To compromise the security of a leaf node, an attacker must crack all the links associated with that node. The probability of data leakage of leaf nodes Pleafi is:

(12) Pleafi=PlostJ−1×PlostJirec×Plostnumfather=PlostJ+Jirec

where i denotes any leaf node, Plost denotes the probability of one link that belongs to this node is broken, Jirec denotes the number of data fragments received by this node, numfather denotes the number of parent nodes of this node and always equal to 1.

The link of the intermediate node of DADIE includes its links to the child nodes and the links that receive data fragments from other leaf nodes, so the probability of data leakage of the intermediate node Pinteri is:

(13) Pinteri=Plostnumchild×PlostJirec×Plostnumfather           =Plostnumchild+Jirec+1.

where numchild denotes the number of child nodes of this intermediate node.

The above description analyzes the security of DADIE in one round of data transmission, however, the data transmission topology tree is reconstructed before each round of data transmission, so even if all the data of a node is captured, it does not affect the privacy protection of the whole network.

Communication and energy analysis

In the DADIE algorithm, we mainly consider the communication overhead and energy consumption in three phases: formation of topology, slicing and mixing, and data transmission.

Communications overhead

In forming the topology, the nodes in the network start from the BS and join the aggregation tree sequentially and establish a secure transmission channel between the node and the parent node. Assuming that the total number of nodes in the network is N; the number of neighboring nodes for each node is Nne; the average number of times each node replies to other node’s invitation messages is Nrav; the average number of invitation messages sent per node is Niav, the invitation message is the same length as the reply message, both are ρ¯bit. Therefore, the communication overhead TSC in forming the topology is:

(14) TSC=∑i=1N(ρ¯*(Nnei+Nrav+Niav)).

At the slicing and mixing stage, DADIE performs a slice-and-mix operation at the leaf nodes, where each leaf node, after it finishes sensing its own sensor data, splits its own sensory data into J segments of the same size. The leaf node keeps one fragment for itself and transmits the remaining J-1 fragments randomly to neighboring nodes other than the parent, and each neighboring node will only be transmitted one fragment. After a node receives a data fragment from a neighboring leaf node, it mixes its own sensory data with the received data fragment as its transmission data. Suppose, in the tree structure constructed by DADIE, the proportion of leaf nodes is α, the size of the sensory data of each node is consistently D, and the communication overhead in the slicing and mixing phase is:

(15) TSM=N∗α∗(J−1J)∗D.

In the data transmission phase, the data transmission starts from the leaf nodes and each leaf node transmits the transmission data to the parent node. The parent node receives the data from the child nodes, mixes it with its own originally transmitted data, and transmits the mixing result to the parent node. This operation is performed until the data is transmitted to the BS. Assume that Dr denotes the total amount of data received by each node as data fragments, Dchild denotes the total amount of data received by each intermediate node from the child nodes,then, the communication overhead TCOM in the data transmission phase is:

(16) TCOM=∑i=1N∗(1−α)⁡(D+Dri+Dchildi).

In summary, the communication overhead of the DADIE algorithm TDADIE is:

(17) TDADIE=TSC+TSM+TCOM.

Energy consumption

In DADIE, the energy consumption model focuses on the radio energy consumption of sensor nodes during data transmission and reception. The model adopts a first-order approach, which means that it considers the energy consumed solely during the communication process and neglects energy losses associated with computation and storage. The energy consumption is calculated separately for the three phases of forming topology, slicing and mixing, and data transmission.

In the stage of forming topology, it is assumed that the total number of nodes in the network is N, the number of neighboring nodes of each node is Nne, the average number of times each node replies to the invitation messages of other nodes is Nrav, the average number of invitations sent by each node is Niav, and the length of invitation messages and replies are the same, ρ¯bit. The energy consumption ETOPs at the sending end is:

(18) ETOPs=∑i=1N∑j=1Nnei(ρ¯*(Een+β(Lj)))               + N*∑i=1Nrav(ρ¯*(Een+β(Li)))               + N*∑i=1Niav(ρ¯*(Een+β(Li))).

The energy consumption ETOPr at the receiving end is:

(19) ETOPr=TSC∗Een.

Therefore, the energy consumption ETOP in the formation topology phase is:

(20) ETOP=ETOPs+ETOPr.

In the slicing and mixing phase, it is assumed that, in the tree structure constructed by DADIE, the proportion of leaf nodes is α, the size of the sensory data of each node is consistently D, and the energy consumption at the sending end is:

(21) ESMs=∑i=1N*α∑j=1J−1(DJ*(Een+β(Lj))).

The energy consumption ESMr at the receiving end is:

(22) ESMr=N∗α∗J−1J∗D∗Een.

Therefore, the energy consumption ESM in the slice mixing stage is:

(23) ESM=ESMs+ESMr.

In the data transmission phase, it is assumed that Dr denotes the total amount of data received by each node in the data fragment and Dchild denotes the total amount of data received by each intermediate node from the child node, the energy consumption ECOMs at the sending end is:

(24) ECOMs=∑i=1N*α((DJ+Dri)*(Een+β(Li)))                  +∑i=1N*(1−α)((D+Dri+Dchildi)*(Een+β(Li)))

where Li denotes the distance of node i from its parent. The energy consumption ECOMr at the receiving end is:

(25) ECOMr=TCOM∗Een.

Therefore, the energy consumption ECOM in the data transmission phase is:

(26) ECOM=ECOMs+ECOMr.

Simulation and analysis

In the previous section, we discussed the significance of metrics such as energy consumption, network lifetime, and security when evaluating data aggregation algorithms for wireless sensor networks. We also provided a theoretical analysis of the DADIE algorithm in relation to these three perspectives. In this section, we will conduct simulation experiments to compare the privacy preservation, communication overhead, energy consumption, and network lifetime of three algorithms: PECDA (Wang et al., 2018), EPDA (Zhou et al., 2019), and DADIE.

Simulation environment configuration

In real-world production scenarios, wireless sensor networks often consist of a large number of nodes deployed over a wide geographical area. To simulate such network structures, we conducted simulation experiments using MATLAB 2017a. We utilized the WNSs scenario and sensor node parameters provided by the Intel Berkeley Research Lab, as referenced in Samuel (2004). In our simulation, we deployed N = 300 sensor nodes within a 100 m × 100 m area denoted as G. Each sensor node, including the BS, had a sensing radius R of 10 m. The initial energy EN of the sensor nodes varied between 0.5 J and 2 J. Based on the specifications outlined in Kavitha et al. (2023), we assumed that the Mica2Dot sensors collected network information and sensory data (e.g., temperature, humidity, voltage, light) every 31 s. The sensory packet size for each node was set to 8,000 bits, while the information packet size was 40 bits. The energy consumption per bit for sending or receiving data was 50 nJ, and the energy consumption for processing data was 5 nJ/bit. During the experiments, we investigated the impact of varying the value of J, representing the number of fragments, using the values 3, 4, 5, and 6. After analyzing the results, we determined that the network achieved the lowest energy consumption while maintaining security when J = 3. To ensure the reliability of our findings, we conducted 20 independent experiments, each with a unique distribution of nodes within the network. The reported experimental data represents the average results obtained from these 20 experimental runs.

To provide a comprehensive overview of the simulation settings, we present the complete parameter configurations in Table 3. This table includes information such as the number of nodes, the geographical area, the sensing radius, the initial energy levels, the sizes of sensory and information packets, the energy consumption values, and the chosen value of J.

Table 3 Simulation system parameters setting.

Parameter	Value	
N	300	
G	100 m * 100 m	
R	10 m	
EN	[0.5 2] J	
DNIDS	40 bits	
D	8,000 bits	
J	3	
Een	50 nJ/bit	
Ede	5 nJ/bit	
εFS	10 pJ/bit/m2	
εAMP	0.0013 pJ/bit/m4	

Privacy-preserving

A comparison of the simulation results for the proportion of node data leakage is presented in Fig. 6, depicting the performance of the EPDA, PECDA, and DADIE algorithms. The horizontal axis represents the probability of communication links between nodes being compromised, while the vertical axis represents the average proportion of cracked links in the wireless sensor network across 20 independent experiments. This measurement reflects the increase in the proportion of cracked links as the probability of link loss for a node rises.

Figure 6 Comparison of privacy preserving of EPDA, PECDA and DADIE.

The EPDA algorithm aims to reduce the number of leaf nodes in the network, which in turn reduces the number of data fragments. However, this algorithm organizes the network topology in a chain structure, resulting in a higher number of intermediate nodes and fewer child nodes for each intermediate node. As intermediate nodes lack data slicing operations, there is an increased likelihood of data leakage at these nodes compared to leaf nodes. This higher probability of data leakage at intermediate nodes can potentially compromise the privacy of the entire network. On the other hand, both the PECDA and DADIE algorithms have a higher number of leaf nodes and a lower number of intermediate nodes. This results in more child nodes participating in data slice-and-mix operations, reducing the probability of data leakage. The PECDA algorithm, however, considers the temporal correlation of sensory data during continuous transmission. This means that some leaf nodes may transmit data fragments to the same destination node in different transmission rounds, increasing the probability of data leakage from these nodes. In contrast, the DADIE algorithm dynamically rebuilds the aggregation tree before each round of data transmission. This ensures that leaf nodes randomly select sliced transmission object nodes from neighboring nodes in each transmission round. This random selection process enhances privacy preservation by reducing the predictability of data transmission paths. As a result, the DADIE algorithm demonstrates slightly better privacy preservation compared to PECDA. The percentage curve of links cracked for the DADIE algorithm shows a flat upward trend and is lower than that of PECDA and EPDA. This indicates that DADIE provides improved privacy preservation performance compared to the other two algorithms.

Overall, experimental results demonstrate that the DADIE algorithm is more efficient and robust in preserving privacy in wireless sensor networks. Its dynamic tree rebuilding and random selection of transmission object nodes contribute to enhanced privacy protection compared to PECDA and EPDA.

Communication overhead and energy consumption

In wireless sensor networks, the communication overhead and energy consumption are influenced by several factors, including the number of nodes, the number of slices, and the network topology. To compare the communication overhead and energy consumption of the three algorithms (PECDA, EPDA, and DADIE), we conducted experiments while controlling the number of sensor nodes and the number of leaf node slices.

In our experiments, we varied the number of nodes in the network from 300 to 1,000, with increments of 100 for each experimental run. This allowed us to observe the impact of network size on communication overhead and energy consumption. Additionally, we varied the number of leaf node fragments from 2 to 6 while keeping the network size fixed at 300 nodes. This enabled us to analyze the effect of the number of slices on the performance of the algorithms.

Figure 6 shows the results of the experimental comparison. The EPDA involves organizing the subtrees of intermediate nodes in the sensor network’s tree topology into a chain structure. This approach reduces the number of leaf nodes in the network, thereby minimizing the number of fragment transmissions and achieving the goal of reducing communication overhead and conserving energy. However, while reducing the number of leaf nodes, EPDA increases the number of intermediate nodes, leading to more data transmission between them and increasing the number of layers in the topology tree. As a result, data transmission to the BS requires more hops, resulting in a greater communication overhead and energy consumption in the network compared to the PECDA and DADIE algorithms.

PECDA algorithm considers the temporal correlation of data during continuous data sensing transmission in WSNs. Specifically, when a leaf node transmits sensory data, PECDA checks whether the difference between the sensed data of two consecutive rounds of data transmission is less than a threshold. If the difference is below the threshold, the node broadcasts a marker signal (1 bit) instead of transmitting the sensed data in the current round. This signal instructs the receiving node to use the base data (typically the data transmitted in the previous round) as the data transmitted in the current round. This approach reduces communication overhead and energy consumption in the network. As a result, when the number of sensor nodes in the network is small, DADIE’s performance is comparable to PECDA. However, as the number of nodes in the network increases, DADIE’s communication overhead and energy consumption gradually become slightly higher than that of PECDA. This can be attributed to the fact that intermediate nodes consume more energy in wireless sensor networks. To address this issue, the DADIE algorithm selects intermediate nodes based on both the transmission distance and the initial energy level, aiming to extend the network’s lifetime. By maximizing the utilization efficiency of intermediate nodes and extending their lifetime, DADIE reduces the number of layers in the tree topology. Consequently, the number of data transmissions from intermediate nodes to other intermediate nodes is significantly reduced, resulting in a notable decrease in communication overhead and energy consumption in the network.

As illustrated in Figs. 7A and 7C, When the number of node slices is constant, DADIE exhibits lower communication overhead, less energy consumption and better network performance as the number of nodes in the network increases. Furthermore, as depicted in Figs. 7B and 7D, when the number of nodes in the network is fixed, the communication overhead and energy consumption of DADIE are always lower than PECDA and EPDA, regardless of the number of slices used.

Figure 7 Comparison of communication overhead and energy consumption of EPDA, PECDA and DADIE algorithms with different number of slices and nodes: (A) communication overhead of the network with different number of nodes; (B) communication overhead of the network with different number of fragments; (C) energy consumption of the network with different number of nodes; (D) energy consumption of the network with different number of fragments.

Network lifetime

To compare the network lifetime of the DADIE, PECDA, and EPDA algorithms, the simulation system conducts 100 rounds of data aggregation operations for each algorithm under the same conditions. The node extinction rate is recorded for each aggregation algorithm in different rounds.

Figure 8 depicts a wireless sensor network comprising 300 randomly distributed nodes within a 100 m × 100 m area, with initial energy values ranging between [0.5, 2] J. The EPDA, PECDA, and DADIE algorithms are compared based on the number of node deaths in the network during varying cycles of data aggregation. As the number of data transmission cycles increases, both the EPDA and PECDA algorithms exhibit a rapid increase in node deaths. However, due to PECDA’s consideration of the temporal correlation of sensed data during continuous data sensing, it achieves a lower node death rate compared to EPDA, effectively conserving network energy. EPDA forms subtrees of intermediate nodes in the network topology, resulting in a chain-like structure that increases the number of hops required for node data transmission to reach the BS. Consequently, this places a heavier load on intermediate nodes near the BS, leading to a rapid increase in the number of node deaths. Conversely, in the DADIE algorithm, the integration of distance and energy allows for the involvement of intermediate nodes with higher initial energy levels and smaller distances from the BS in a greater number of data aggregation and transmission operations, while leaf nodes with lower energy levels participate in fewer operations. As a result, the number of node deaths in the DADIE algorithm exhibits a smoother and slower growth pattern.

Figure 8 Number of node deaths for different rounds of EPDA, PECDA and DADIE continuous data transmission.

As shown in Fig. 8, the node death rate of the DADIE algorithm is approximately 10–30% lower than that of EPDA and PECDA. This indicates that DADIE effectively achieves a balanced network energy load in WSNs, prolonging their lifetime and exhibiting excellent stability.

Conclusion

This article proposes a data aggregation algorithm called DADIE for WSNs with nodes having different initial energy levels. DADIE consists of three phases: topology formation, slicing and mixing, and data transmission. In the topology formation phase, comprehensive consideration is given to the distance between nodes and their initial energy levels. Nodes that are closer to the BS and have higher initial energy levels are selected as intermediate nodes to achieve a balanced energy load in the network. This approach takes into account the fact that nodes closer to the BS tend to have higher energy consumption. During the slicing and mixing phase, slicing and transmission operations are performed only at the leaf nodes to reduce the communication load on the network while maintaining node privacy. This ensures efficient data aggregation while preserving privacy. In the data transmission phase, secure data transmission channels are established in advance between neighboring nodes. This eliminates the need for multiple encryption and decryption processes during data transmission, thereby reducing energy consumption. Additionally, before each round of data transmission, DADIE reconstructs a new aggregation tree to balance the network’s energy consumption and slow down the node death rate. Simulation experiments are conducted in this study to compare the performance of DADIE with existing algorithms, namely PECDA and EPDA. The results demonstrate that DADIE excels in terms of privacy preservation, communication overhead, energy consumption, and network lifetime.

However, there are still some limitations in this study. Firstly, the proposed DADIE algorithm does not fully consider additional parameters that can impact data aggregation in wireless sensor networks, such as the sensor battery performance and node mobility in data collection, and other relevant factors. These parameters can play a significant role in determining the optimal aggregation tree. Not accounting for these factors may limit the algorithm’s performance in real-world scenarios. Secondly, the simulation experiments were conducted using MATLAB under ideal conditions, which are mathematical simulations rather than real-world experiments in practical scenarios. The ideal conditions may not accurately reflect the complexities and uncertainties present in real-world wireless sensor networks. Therefore, the performance evaluation of the DADIE algorithm based solely on these simulations may not provide a comprehensive understanding of its effectiveness in practical scenarios. It is important to further investigate and refine the DADIE algorithm by considering additional parameters and conducting real-world experiments to validate its performance. These efforts would enhance the algorithm’s applicability and provide more reliable results.

In the future, our focus will be on optimizing the algorithm for forming the topology in DADIE. We aim to consider more influential parameters that can help reduce communication overhead and energy consumption in data aggregation from the perspective of network topology. By incorporating these parameters into the algorithm, we anticipate achieving more efficient and effective data aggregation in wireless sensor networks. Furthermore, we are committed to enhancing network security within the algorithm for forming the wireless sensor network topology in DADIE. We will work towards strengthening the security measures and ensuring that the algorithm is robust and scientifically grounded, thereby mitigating potential vulnerabilities and threats. In addition, we plan to conduct real-world experiments with multiple scenarios and different sensors. With the results of real-world experiments, we can verify the performance of the algorithm and evaluate its effectiveness.

Supplemental Information

Supplemental Information 1 Code for mathematical simulation of the DADIE data aggregation algorithm.

Supplemental Information 2 Code for mathematical simulation of the EPDA data aggregation algorithm.

Supplemental Information 3 Code for mathematical simulation of the PECDA data aggregation algorithm.

Supplemental Information 4 Parameter setting code for the simulation experiment, including the number of nodes, energy consumption, node distribution area, etc.

Supplemental Information 5 Code for randomly generating a fixed number of nodes in a fixed area.

Additional Information and Declarations

Competing Interests

Author Contributions

Data Availability

The authors declare that they have no competing interests.

Zhenpeng Liu conceived and designed the experiments, performed the experiments, authored or reviewed drafts of the article, and approved the final draft.

Jialiang Zhang conceived and designed the experiments, performed the experiments, analyzed the data, performed the computation work, prepared figures and/or tables, and approved the final draft.

Yi Liu conceived and designed the experiments, performed the experiments, authored or reviewed drafts of the article, and approved the final draft.

Fan Feng performed the computation work, prepared figures and/or tables, and approved the final draft.

Yifan Liu performed the experiments, analyzed the data, performed the computation work, prepared figures and/or tables, authored or reviewed drafts of the article, and approved the final draft.

The following information was supplied regarding data availability:

The code for the comparison experiment, the parameter conditions of the sensor networks, the code implementations for DADIE, EPDA, and PECDA, slice mixing operation, and the code implementations for the data transfer available in the Supplemental Files.

The data used in the comparison experiments of this study was not real data, they were set up to be false for conducting the comparison experiments.

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
