# Peer review of "Data aggregation algorithm for wireless sensor networks with different initial energy of nodes"

_PeerJ Computer Science, doi:10.7717/peerj-cs.1932_

## Round 0.1 · original submission · Major Revisions

Dear authors,

Thank you for submitting your article. The reviewers’ comments are now available. Your article has not been recommended for publication in its current form. However, we encourage you to address the reviewers' concerns and criticisms; particularly regarding readability, quality, experimental design and validity, and resubmit your article once you have updated it accordingly.

Reviewer 2 has requested that you cite specific references. You may add them if you believe they are especially relevant and useful. However, I do not expect you to include these citations, and if you do not include them, this will not influence my decision.

Best wishes,

**Language Note:** The review process has identified that the English language must be improved. PeerJ can provide language editing services - please contact us at copyediting@peerj.com for pricing (be sure to provide your manuscript number and title). Alternatively, you should make your own arrangements to improve the language quality and provide details in your response letter. – PeerJ Staff

Reviewer 1 ·

Basic reporting

1-It is a known fact that there are many algorithms developed to collect data in wireless sensor networks. However, the authors proposed a new algorithm called DADIE to create the initial energy levels, which are considered the basis of these algorithms. I believe that this problem, to which the authors propose a solution, is extremely important.
2-However, in the introduction part of the article, it should be stated what problems will arise from assuming an initial energy level of network devices in wireless sensor networks. Many related works in the literature should be added here.
3- "We demonstrate the feasibility of DADIE using simulation to model comparative experiments." They use the expression in the main contributions section. However, it should also mention which simulation tool they used and what kind of performance it achieved. Additionally, the main contributions section should be expanded.
4-Authors have left almost no work in related works since 2023. This part should definitely be developed and these studies should not be summarized simply. These studies should be examined with their positive and negative aspects.

Experimental design

5-The authors suggest that it would be a better approach if the DADIE algorithm could be compared with the performance rates of algorithms that accept the initial energy level of sensors that measure parameters such as air humidity, temperature or wind speed in wireless sensor networks.
6-In the graph shown in Figure 5, the performances of PECDA and DADIE are seen close to each other. Is this graph obtained as the average of 20 independent runs?

Validity of the findings

7-Detailed information should be given about the main limitations of this study.
8-Information should be given about future studies planned.
9-The conclusion part of the study should be improved.

Reviewer 2 ·

Basic reporting

The manuscript titled "Data Aggregation Algorithm for Wireless Sensor Networks with Different Initial Energy of Nodes" by Zhenpeng Liu et al. proposes a novel algorithm, DADIE (Data Aggregation with Different Initial Energy), aimed at enhancing energy efficiency and security in wireless sensor networks with nodes having varying initial energy levels. The authors address the challenge of premature node death and energy imbalances by considering transmission distance and initial energy levels in network topology formation, limiting the number of child nodes, and reconstructing the aggregation tree before each data transmission round. Hence, the authors need to attend to the following Comments for improvement of their study:
1. The manuscript is generally well-structured, but certain sections could benefit more clarity. Ensuring that each section flows logically into the next and delineates different aspects of the research would enhance readability.
2. The methods and algorithms are well described, but including more detailed explanations or examples for readers unfamiliar with the specific technologies or techniques might be helpful.
3. The results are interesting, but the paper could benefit from a more comprehensive analysis. Consider including a broader range of test scenarios or comparative analysis with other existing solutions to strengthen the findings.
4. The discussion effectively highlights the significance of the research, but it could be expanded to articulate better the proposed algorithm's potential practical applications and limitations.
5. Ensure all figures and tables are clear, well-labelled, and relevant to the manuscript's content. Consider adding more graphical representations of the data to enhance understanding.
6. It is observed that the authors cited only three publications in 2021 and 2022 and only 1 in 2023. Citing recent literature has several advantages for both authors and journals. It can help authors establish their credibility, demonstrate their research's relevance, and help avoid plagiarism. In the same way, it assists journals in increasing their visibility, improving their reputation, increasing their citation rates, and meeting reader expectations. For this reason, I have suggested some recent literature from 2021 to 2023 relating to the study that you must read, cite and reference in your article.

a. Zhou, G., Zhang, R., & Huang, S. (2021). Generalized Buffering Algorithm. IEEE Access, 9, 27140-27157. doi: 10.1109/ACCESS.2021.3057719
b. K. Ma et al., "Reliability-Constrained Throughput Optimization of Industrial Wireless Sensor Networks With Energy Harvesting Relay," in IEEE Internet of Things Journal, vol. 8, no. 17, pp. 13343-13354, 1 Sept.1, 2021, doi: 10.1109/JIOT.2021.3065966.
c. G. Liu, "Data Collection in MI-Assisted Wireless Powered Underground Sensor Networks: Directions, Recent Advances, and Challenges," in IEEE Communications Magazine, vol. 59, no. 4, pp. 132-138, April 2021, doi: 10.1109/MCOM.001.2000921
d. Cao, K., Ding, H., Li, W., Lv, L., Gao, M., Gong, F.,... Wang, B. (2022). On the Ergodic Secrecy Capacity of Intelligent Reflecting Surface Aided Wireless Powered Communication Systems. IEEE Wireless Communications Letters, PP, 1. doi: 10.1109/LWC.2022.3199593
e. Wu, H., Jin, S., & Yue, W. (2022). Pricing Policy for a Dynamic Spectrum Allocation Scheme with Batch Requests and Impatient Packets in Cognitive Radio Networks. Journal of Systems Science and Systems Engineering, 31(2), 133-149. doi: 10.1007/s11518-022-5521-0
f. Chung, K. L., Tian, H., Wang, S., Feng, B., & Lai, G. (2022). Miniaturization of microwave planar circuits using composite microstrip/coplanar-waveguide transmission lines. Alexandria Engineering Journal, 61(11), 8933-8942. doi: https://doi.org/10.1016/j.aej.2022.02.027
g. Jiang, Y., & Li, X. (2022). Broadband cancellation method in an adaptive co-site interference cancellation system. International journal of electronics, 109(5), 854-874. doi: 10.1080/00207217.2021.1941295
h. Shuai Wang, Hao Sheng, Da Yang, Yang Zhang,Yubin Wu,Sizhe Wang. (2022) Extendable Multiple Nodes Recurrent Tracking Framework with RTU++. IEEE Transactions on Image Processing. Doi: 10.1109/TIP.2022.3192706
i. Mao, Y., Sun, R., Wang, J., Cheng, Q., Kiong, L. C.,... Ochieng, W. Y. (2022). New time-differenced carrier phase approach to GNSS/INS integration. GPS Solutions, 26(4), 122. doi: 10.1007/s10291-022-01314-3
j. Mao Y, Zhu Y, Tang Z, Chen Z. A Novel Airspace Planning Algorithm for Cooperative Target Localization. Electronics. 2022; 11(18):2950. https://doi.org/10.3390/electronics11182950
k. Wang, Y., Sun, R., Cheng, Q., & Ochieng, W. Y. (2023). Measurement quality control aided multi-sensor system for improved vehicle navigation in urban areas. IEEE Transactions on Industrial Electronics. doi: 10.1109/TIE.2023.3288188
l. Li, S., Chen, J., Peng, W., Shi, X., & Bu, W. (2023). A vehicle detection method based on disparity segmentation. Multimedia Tools and Applications, 82(13), 19643-19655. doi: 10.1007/s11042-023-14360-x
m. Bo, C., Jiangping, H., & Bijoy, G. (2023). Finite-Time Observer Based Tracking Control of Heterogeneous Multi-AUV Systems with Partial Measurements and Intermittent Communication. SCIENCE CHINA Information Sciences. doi: https://doi.org/10.1007/s11432-023-3903-6
n. Jiang, Y., Liu, S., Li, M., Zhao, N., & Wu, M. (2022). A new adaptive co-site broadband interference cancellation method with auxiliary channel. Digital Communications and Networks. doi: https://doi.org/10.1016/j.dcan.2022.10.025
o. Mi, C., Huang, S., Zhang, Y., Zhang, Z., & Postolache, O. (2022). Design and Implementation of 3-D Measurement Method for Container Handling Target. Journal of Marine Science and Engineering , 10(12), 1961. doi: https://doi.org/10.3390/jmse10121961
p. Zheng, W., Gong, G., Tian, J., Lu, S., Wang, R., Yin, Z.,... Yin, L. (2023). Design of a Modified Transformer Architecture Based on Relative Position Coding. International Journal of Computational Intelligence Systems, 16(1), 168. doi: 10.1007/s44196-023-00345-z
q. Guo, R., Liu, H., & Liu, D. (2023). When Deep Learning-Based Soft Sensors Encounter Reliability Challenges: A Practical Knowledge-Guided Adversarial Attack and Its Defense. IEEE Transactions on Industrial Informatics. doi: 10.1109/TII.2023.3297663
7. While the technical content is strong, the manuscript could benefit from a thorough language and grammar check to enhance its professional quality.
8. Strengthen the concluding section by summarizing key findings more explicitly and suggesting potential areas for future research based on the study's outcomes.

Experimental design

The manuscript's experimental design is innovative in addressing the complexity of varying initial energy levels in sensor nodes, a critical aspect of wireless sensor network research. It relies on simulation-based approaches for initial testing, although the paper could explore the limitations of this method compared to real-world applications. The chosen metrics for evaluation, including energy efficiency, network lifespan, and security, are pertinent, yet a more detailed rationale for these choices is suggested. The algorithmic phases are well described, but further details or illustrations could aid understanding. Enhancing reproducibility through detailed simulation environment descriptions, discussing the practical significance of results, and elaborating on security improvements would strengthen the paper.

Validity of the findings

The systematic approach and thorough experimental setup used in the study appear to support the validity of the findings in the manuscript "Data Aggregation Algorithm for Wireless Sensor Networks with Different Initial Energy of Nodes." By comparing the new DADIE algorithm to other algorithms based on important factors like energy efficiency, network lifespan, and security, this study builds a strong foundation for judging how well the new algorithm works. However, real-world testing or more varied simulation scenarios to supplement the current simulation-based results would further strengthen the validity of these findings. Additionally, a deeper exploration of the algorithm's scalability and adaptability to different types of wireless sensor networks could enhance the comprehensiveness of the findings.

Additional comments

No comment

---

## Round 0.2 · accepted · Accept

Dear authors,

Thank you for the revision. I confirm that the paper is improved and addresses the reviewers' concerns regarding the basic reporting, experimental design and validity of the results. With this revision, your paper is now acceptable for publication.

Best wishes,

Reviewer 1 ·

Basic reporting

It can be seen in the article that all the changes I requested from the authors in the previous revision were made. Additionally, the authors answered many of my questions in the revision.

Experimental design

It can be seen in the article that all the changes I requested from the authors in the previous revision were made. Additionally, the authors answered many of my questions in the revision.

Validity of the findings

It can be seen in the article that all the changes I requested from the authors in the previous revision were made. Additionally, the authors answered many of my questions in the revision.

Additional comments

It can be seen in the article that all the changes I requested from the authors in the previous revision were made. Additionally, the authors answered many of my questions in the revision.

Reviewer 2 ·

Basic reporting

Following the comprehensive revisions made to your manuscript in line with the provided comments, I am pleased to inform you that your paper is now recommended for acceptance. Your efforts in addressing the concerns and incorporating the suggested changes have significantly improved the quality of your work, aligning it with the publication standards. Congratulations, and thank you for your diligent revisions.

Experimental design

No comment

Validity of the findings

No comment

Additional comments

No comment